

**Development of a new gas flaring emission data set for southern West Africa**
Konrad Deetz, Bernhard Vogel
Karlsruhe Institute of Technology (KIT), Institute of Meteorology and Climate Research – Troposphere Research
(IMK-TRO), Hermann-von-Helmholtz 1, 76344 Eggenstein-Leopoldshafen
**HIGHLIGHTS**
- Development of a new gas flaring emission parameterization for air pollution modeling.
- Combination of remote sensing observation and physical based combustion calculation.
- Application to the significant gas flaring region southern West Africa.
- Comprehensive assessing of the parameterization uncertainties.
- Comparison with existing gas flaring emission inventories.
**Keywords:**
Gas flaring
Emission parameterization
Emission uncertainty
Pollution modeling
Carbon dioxide

**ABSTRACT**

A new gas flaring emission parameterization has been developed which combines remote sensing
observations using VIIRS nighttime data with combustion equations. The parameterization has been
applied to southern West Africa, including the Niger Delta as a region which is highly exposed to gas
flaring. Two two-month datasets for June-July 2014 and 2015 were created. The parameterization
delivers emissions of $CO$, $CO_2$, $NO$ and $NO_2$. A flaring climatology for both time periods has been
derived. The uncertainties owing to cloud cover, parameter selection, natural gas composition and
the interannual differences are assessed. Largest uncertainties in the emission estimation are linked
to the parameter selection. By using remote sensing cloud cover observations, a correction factor for
the climatology was established to consider the effect of flares masked by clouds. It can be shown
that the flaring emissions in SWA have significantly decreased by 30% from 2014 to 2015. Existing
emission inventories were used for validation. $CO_2$ emissions with the estimated uncertainty in
brackets of 8 ($^{12}/_2$) Tg y$^{-1}$ for 2014 and 5 ($^{7}/_1$) Tg y$^{-1}$ for 2015 are derived. The flaring emission
estimation within this study for June-July 2014 is in the same order of magnitude compared to
existing emission inventories. For the same period in 2015 the emission estimation is one order of
magnitude smaller in comparison to existing inventories. The deviations might be attributed to
uncertainties in the derived flare gas flow rate, the decreasing trend in gas flaring or inconsistent
emission sector definitions. The parameterization source code is available as a package of R scripts.

**1. Introduction**

Gas flaring is a globally used method to dispose flammable, toxic or corrosive vapors to less reactive
compounds at oil production sites and refineries. In regions of insufficient transportation
infrastructure or missing consumers, flaring is also commonly applied.



CDIAC (2015a) estimated the global gas flaring emission of carbon dioxide to 267.7 million tons
(0.83% of total emissions) in 2008. Flaring and venting of gas significantly contributes to the
greenhouse gas emissions and therefore to the global climate change. The five countries with the
highest flaring amount in billion cubic meter (bcm) are Russia (35), Nigeria (15), Iran (10), Iraq (10)
and USA (5) (World Bank, 2012).
In recent time, especially with the development of remote sensing observation techniques (e.g.
Elvidge et al. (1997, 2013)), emissions from gas flaring moved in the focus of atmospheric research
involving the efforts in reducing the pollution and the waste of resources. The World Bank led the
initiatives "Global Gas Flaring Reduction Partnership" (GGFR) and "Zero Routine Flaring by 2030" to
promote the efficient use of flare gas.
Instead of relying on national statistics of gas production and consumption for estimating the flaring
amount, remote sensing techniques can estimate the flaring amount directly via multispectral data
(Elvidge et al., 2013). Elvidge et al. (2009) developed a 15 year dataset of global and national gas
flaring efficiency from 1994 to 2008 by using data from the Defense Meteorological Satellite Program
(DMSP). Doumbia et al. (2014) combined DMSP with emission factors for flaring, to estimate the
flaring emissions for SWA. The satellite product Visible Infrared Imaging Radiometer Suite (VIIRS)
Nightfire (Elvidge et al., 2013), which is free available as "VIIRS Nightfire Prerun V2.1 Flares only"
(VIIRS, 2015) (VNP hereafter), is now the most widely used product to derive flaring emissions from
satellite imagery. By using VNP, Zhang et al. (2015) estimated the methane consumption and the
release of $CO_2$ from gas flaring for the northern U.S. which agree with field data within an uncertainty
range of $\pm50\%$.
Also in the second largest flaring country Nigeria, the awareness of gas flaring increases. On
gasflaretracker.ng the attention of the government, industry and society is called to the flaring
problem by interactive maps of flare infrastructure, amounts and costs. The implications of gas
flaring in Nigeria are far-reaching. It influences the environment by noise and deterioration of the air
quality (Osuji and Avwiri, 2005). Nwankwo and Ogagarue (2011) have measured higher
concentrations of heavy metals in surface water of a gas flared environment in Delta State Nigeria.
Adverse ecological and bacterial spectrum modifications by gas flaring are indicated by Nwaugo et
al. (2006). Gas flaring also causes acid rain which causes economic burden via rapid corrosion of zinc
roofs (Ekpoh and Obia, 2010) and causes retardation in crop growth owing to high temperatures
(Dung et al., 2008).
The project DACCIWA (Dynamics-aerosol-cloud interactions in West Africa, Knippertz et al. (2015))
investigates the influence of anthropogenic and natural emissions on the atmospheric composition
over SWA, including the flaring hotspot Nigeria, to examine the meteorological and socio-economic
effects. To consider the SWA gas flaring emissions (e.g. in an atmospheric model), this study presents
a method to derive emission fluxes by combining the state of the art flaring detection VNP and the
combustion equations of Ismail and Umukoro (2014) which does not use emission factors. The new
parameterization is robust and easy to apply to new research questions according flexibility in the
spatiotemporal resolution.
The parameterization is presented in Section 2. Results of the application to SWA, including the
spatial distribution of gas flaring, the emission estimation and the uncertainty assessment are
investigated in Section 3. Section 4 places the emission estimates in the context of existing
inventories.  The results are summarized and discussed in Section 5.
**2. Parameterization of gas flaring emissions**



The new parameterization for gas flaring presented here, is based on VNP (VIIRS Nightfire Prerun
V2.1 Flares only) and the combustion equations of Ismail and Umukoro (2014) (IU14 hereafter).

## 2.1 Remote sensing identification of gas flares

VIIRS (Visible Infrared Imaging Radiometer Suite) is a scanning radiometer for visible and infrared
light on board the sun-synchronous Suomi National Polar-orbiting Partnership weather satellite
(Suomi-NPP) (NASA, 2016). It can detect combustion sources at night (e.g. bush fires or gas flares) by
spectral band M10. To confirm these sources and to eliminate noise, the Day/Night Band (DNB), M7,
M8 and M12 are used in addition. By fitting these measured spectra to the Planck radiation curve,
background and source temperatures can be deduced. VNP is filtered to include only detections with
temperatures between 1600 K and 2000 K, which is believed to be an adequate estimation for
average gas flares. Up to now no atmospheric correction is done (VIIRS, 2015).
The data is freely available as daily data from March 2014 to present. The files include among others
the location of the flares, source temperature $T_s$, radiant heat $H$ and time of observation.
For this study we have decided for a two month period of observation. This allows a compilation of a
flaring climatology in terms of the locations and emissions and a robust estimation of uncertainty
owing to cloud coverage and other parameters that have to be prescribed for IU14. We have
selected the month June and July because the gas flaring emission dataset will be used within the
regional online-coupled chemistry model COSMO-ART (Vogel et al., 2009) during the measurement
campaignof the project DACCIWA, which takes place in June/July 2016. This campaign includes
airborne, ground based and remote sensing observations of meteorological conditions and air
pollution characteristics. COSMO-ART is one of the forecasting models of the DACCIWA campaign
and delivers spatiotemporal aerosol/chemistry distributions. The data for 2014 and 2015 are used to
allow also for an interannual comparison, to assess the uncertainty owing to changes in flare
processes (e.g. built-up or dismantling, increase or decrease in combustion).
For this study we use location, source temperature and radiant heat for days with sufficient satellite
coverage over the research domain SWA with a focus on the Niger Delta.

## 2.2 Emission estimation method

The principle emission estimation methodology used in this study follows IU14. The gas flaring
emissions are estimated based on combustion equations for incomplete combustion including six
flaring conditions given in Tab. 1. The equations are introduced in detail in IU14 and are therefore
not presented here. This section concentrates on the application of the method of IU14 to the VNP
data and the research domain in SWA.

**Tab.1.** Reaction types for incomplete combustion of flared gas, depending on availability of sulfur in the flared gas and the
temperature in the combustion zone which determines the formation of NO and $NO_2$.

| Reaction type | Sulfur in flared gas | Source temperature (K) | $NO_x$ formation |
|:---:|:---:|:---:|:---:|
| 1 | No | $< 1200$ | no |
| 2 | Yes | $< 1200$ | no |
| 3 | No | $1200 \leq T_s \leq 1600$ | only NO |
| 4 | Yes | $1200 \leq T_s \leq 1600$ | only NO |
| 5 | No | $> 1600$ | NO and $NO_2$ |
| 6 | Yes | $> 1600$ | NO and $NO_2$ |




As input, IU14 needs the natural gas composition $C$ of the fuel input of the flare, the source
temperature $T_s$ (temperature in the combustion zone), and the flare characteristics including
combustion efficiency $\eta$ (1 is complete combustion without Carbon monoxide formation) and
availability of combustion air $\delta$ (above 1 is excess and below 1 is deficiency). In addition we need the
flow rate $F$, the gauge pressure of the fuel gas in the flare $p_g$, and the fraction of total reaction
energy that is radiated $f$. The value for $f$ is estimated by averaging a table of literature values for $f$
given in Guigard et al. (2000). The IU14 input is summarized in Tab. 2.

**Tab.2.** Variables and parameters needed for IU14 or for deriving the fluxes of the air pollutants

| Parameter | Description | Reference | Unit |
|---|---|---|---|
| $C$ | Natural gas composition | Sonibare and Akeredolu (2004) | % |
| $T_s$ | Source temperature | VNP (VIIRS, 2015) | K |
| $\eta$ | Combustion efficiency | 0.8 (IU14) | - |
| $\delta$ | Availability of combustion air | 0.95 (IU14) | - |
| $H$ | Radiant heat | VNP (VIIRS, 2015) | MW |
| $F$ | Flow rate | VNP (VIIRS, 2015), TA-Luft (1986) | $m^3\ s^{-1}$ |
| $p_g$ | Gauge pressure | 34.475 (API, 2007) | kPa |
| $f$ | Fraction of radiated heat | 0.27 (Guigard et al., 2000) | - |


The natural gas composition is taken from Sonibare and Akeredolu (2004). They have measured the
molar composition of Nigerian natural gas in the Niger Delta area for ten gas flow stations. For this
study we have calculated the average over these stations and merged the data according their
number of carbon atoms (Tab. 3). $H_2S$ fraction is rather low because it was detected only in two out
of the ten flow stations.

**Tab.3.** Molar composition of natural gas in Niger Delta (Nigeria) based on the measurements of Sonibare and Akeredolu
(2004), averaged over ten flow station. The hydrocarbons are merged according to the number of C atoms.

| Constituent | Fraction (%) |
|---|---|
| Methan ($CH_4$) | 78.47 |
| Ethan ($C_2H_6$) | 6.16 |
| Propane ($C_3H_8$) | 5.50 |
| Butan ($C_4H_{10}$) | 5.19 |
| Pentane ($C_5H_{12}$) | 3.95 |
| Hexane ($C_6H_{14}$) | 0.36 |
| Carbon dioxide ($CO_2$) | 0.305 |
| Nitrogen ($N_2$) | 0.06 |
| Hydrogen sulfide ($H_2S$) | 0.005 |


The source Temperature $T_s$ is taken from VNP. The combustion efficiency $\eta$ was set to 0.8 and the
availability of combustion air $\delta$ to 0.95. IU14 remarked, that the reaction condition for flaring of
$\eta \gg 0.5$ and $\delta \geq 0.9$ should be the norm in regions, where the effective utilization of this gas is not
available or not economically. Strosher (2000) indicate a combustion efficiency of solution gas at oil-
field battery sites between 0.62 and 0.82, and 0.96 for flaring of natural gas in the open atmosphere
under turbulent conditions. EPA (1985) shows combustion efficiencies between 0.982 and 1 for
measurements on a flare screening facility. Section 3.3.2 will shed light on the uncertainty which
arises from $\eta$ and $\delta$ via a parameter sensitivity study. The authors strongly recommend a careful
selection of $\eta$ and $\delta$ since unrealistic combinations (e.g. higher combustion efficiencies with rather
low availability of combustion air) can lead to negative NO and $NO_2$ emissions.



The flow rate, gauge pressure and fraction of radiated heat are not included in the parameterization
of IU14 but are necessary to derive the mass emission rates which can be used as emission data for
an atmospheric dispersion model.
The flow rate $F$ (m³ s⁻¹) is estimated by Eq. 1 (TA-Luft, 1986)


$$F = M/\left(1.36 \cdot 10^{-3} \left(T_S - 283\right)\right), \qquad (1)$$


where $M$ is the heat flow in MW and $T_S$ the source temperature in K. We assume that the emitted
heat flow $M$ is equal to the total reaction energy of the flare. VNP only detects the energy fraction
that is radiated $H$ and not the total energy $M$. By using the radiant heat $H$ (observed by VNP) and the
factor $f$ (fraction of $H$ to the total reaction energy, Guigard et al., 2000), we estimate $M$ as $H \cdot 1/f$.
For the source temperature $T_S$ we use the VNP observation.
The estimation of the fuel gas density, which is necessary to transform the flow rate $F$ into an
emission, is problematic due to the lack of data concerning the technical setup of the SWA flares. We
assume that the dominating flare type is a low-pressure single point flare. Bader et al. (2011) pointed
out that these flares are the most common flare type for onshore facilities that operate at low
pressure (below 10 psi (69 kPa) above ambient pressure) and API (2007) remarks that most subsonic-
flare seal drums operate in the range from 0 psi to 5 psi (34 kPa). Therefore we have decided for a
gauge pressure $p_g$ of 5 psi (34 kPa) above ambient pressure. Via Eq. 2 we can calculate the fuel gas
density $\rho_f$


$$\rho_f = p_f/\left(R/\left(M_f\, T_a\right)\right), \qquad (2)$$


where $p_f$ is the fuel gas pressure as the sum of ambient pressure (10.1325 kPa, taken as const) and
gauge pressure $p_g$. $R$ is the universal gas constant, $M_f$ the molar mass of the fuel gas and $T_a$ the
ambient temperature (293.15 K, taken as const). Finally, the emission $E$ (kg s⁻¹) of a species $i$ is given
by

$$E_i = \frac{m_i}{m_{total}}\, \rho_f\, F, \qquad (3)$$


where $m_i$ is the mass of the species $i$ and $m_{total}$ the total mass of the fuel gas, both delivered by the
parameterization of IU14.
The combustion calculations within IU14 provide the species water, hydrogen, oxygen, nitrogen,
carbon dioxide, carbon monoxide, sulfur dioxide, nitrogen oxide and nitrogen dioxide. In the
following only CO, $SO_2$, NO and $NO_2$ are considered. However, no black carbon or volatile organic
compounds (VOCs) are considered by IU14, although they are not negligible. Johnson et al. (2011)
estimated the mean black carbon emission for a large-scale flare at a gas plant in Uzbekistan to be
7400 g h⁻¹ and Strosher (1996) measured the concentration of predominant VOCs 5 m above the gas
flare in Alberta with 458.6 mg m⁻³. However, owing to the missing representation of black carbon and
VOCs in IU14, these compounds are not considered in this study.
A flaring emission comparison between several days or averaging over a certain period is problematic
due to small variances in the VNP locations of the flares. This means even the same flare can be
detected on a slightly different position the next day, which makes an emission averaging for every
single flare difficult, especially in intensive flare areas. We bypass the problem by predefining a grid
and allocating the flares to this grid. By using the source code written in R (R Core Team, 2013)



delivered by this study, the user can define the grid size independently. For calculating the average
over several days, the emissions for every single flare per day are calculated and summed up
according to their belonging to a certain grid box. This leads to one big point source per grid box. The
corresponding emissions are then averaged over the time period of interest for every grid box (flare
box herafter). Considering this approach within an atmospheric model, by selecting the same grid
configuration for the flaring emission data and the model, no loss of information occurs.
**3. Results**
3.1 Spatial distribution of gas flaring in SWA
We have selected the two time periods June/July 2014 (TP14) and June/July 2015 (TP15) and omitted
all days without observations or with insufficient data coverage for VNP over SWA. This leads to 58
(48) observations for TP14 (TP15).
In the preparation of this work we have compared the estimated mean locations of the flares of TP14
with the Google Earth imagery (Google Earth, 2014) (not shown). Only the onshore flares are visible
in Google Earth. This visual verification reveals that 72% of the VNP detected onshore flares are
visible in Google Earth. It is very likely that the hit rate is much higher since it is often the case that
the Google Earth image quality is not good enough for verification or the images are not up to date.
This comparison indicates that VNP is a valid method to identify the flares in SWA.
For the following analysis we have calculated the emissions for both time periods on a grid with a
mesh size of 0.25° (28 km) from 10°S to 10°N and from 10°W to 15°E. Fig. 1 emphasizes the areas in
which VNP detects flares only in TP14 (TP15) in red (green) color and in grey the areas with flaring in
both periods.

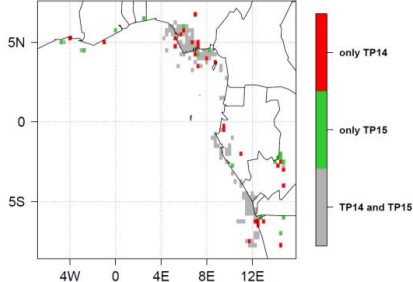

**Fig.1.** Flaring area for TP14 and TP15. Red (green) boxes denote areas with flaring only for TP14 (TP15). For the grey areas,
flaring is detected in both time periods.
Remarkable are the dominating flaring areas in the Niger Delta and the adjacent offshore regions in
the Gulf of Guinea. Also in the coastal region of Gabon, Republic of the Congo, Angola and
sporadically along the coast of Ivory Coast, Ghana and Benin, flaring occurs. By comparing TP14 and
TP15 more red than green areas are visible, especially in southern Nigeria, which indicates a
reduction in the flaring area from 2014 to 2015. A decrease in $CO_2$ from 1994 to 2010, particularly in
the onshore platforms is indicated by Doumbia et al. (2014).
The mean active flare density, as the sum over all detected flares in a box averaged over the time
period, is shown in Fig. 2 for (a) TP14 and (b) TP15.



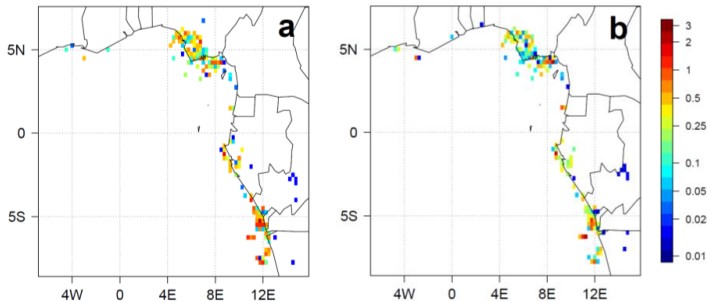

**Fig.2.** Mean active flare density (number of active flares per box) averaged over (a) TP14 and (b) TP15 in logarithmic scale

Fig. 2 shows to a a reduction in the active flare density in TP15 compared to TP14. 72% of the flaring
area which TP14 and TP15 have in common, shows a reduction in TP15 about 48% on average. 28%
of the common flaring area shows an increase in TP15 about 124%. Therefore it seems that the
flaring intensity decreases in TP15 over large areas but simultaneously some flaring hotspots
occurred, which are distributed along the SWA coast (not shown). Fig. 2, together with the variation
of flaring emissions from TP14 to TP15 in Section 3.3.3, indicates the high year to year variations. This
makes the use of past averaged conditions questionable, especially when certain episodes are
studied.

3.2 Emission estimation

For the emission estimation we have used a climatological approach ($E_{clim}$). For every day with valid
data in TP14 and TP15 the emissions for all detected flares are calculated separately and allocated to
the predefined grid. The emissions are summed up in every flare box to have one joined flare per grid
box. Finally the temporal average for every grid box is calculated over TP14 and TP15 respectively.
Therefore all flares, detected in the time period, are active at once with their mean emission
strength. This method has the advantage that most likely all flares in the domain are captured even if
a fraction of them is covered by clouds at certain days. However, this could lead to an emission
overestimation because not all available flares are active at once. This problem of separating
between flares which are not active and flares which are active but covered by clouds and therefore
not visible for VNP is picked up again in Section 3.3.1. Fig. 3 shows the emissions of CO, $SO_2$, NO and
$NO_2$ in t h$^{-1}$ for TP15.



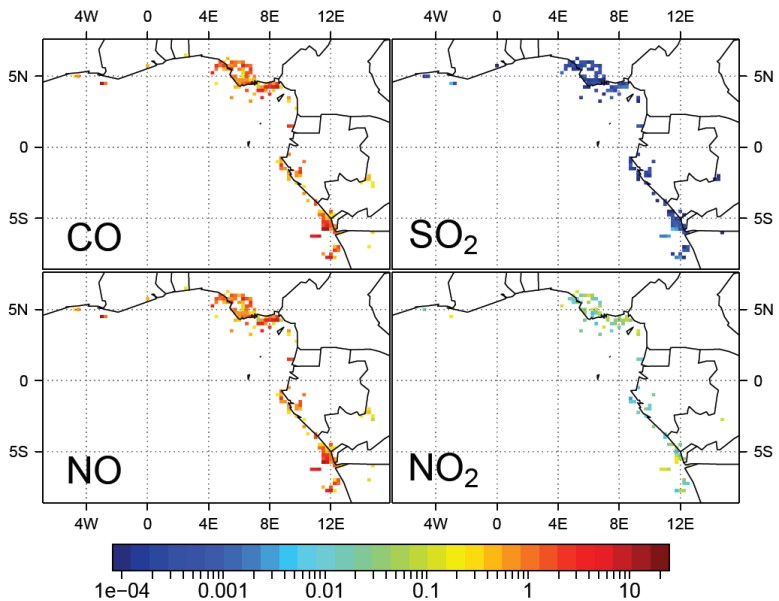

**Fig.3.** Flaring emissions for TP15 within $E_{clim}$ in t h$^{-1}$ for CO, SO$_2$, NO and NO$_2$

Highest emissions are calculated for carbon monoxide, followed by nitrogen oxide and nitrogen dioxide. Sulfur dioxide shows lowest emissions since these emissions do not depend on combustion processes but only on the natural gas composition (see Tab. 3) and the amount of flared gas (IU14). Due to the use of the averaged measurements of Sonibare and Akeredolu (2004), local variations of hydrogen sulfide concentrations in the natural gas cannot be taken into account. Hydrogen sulfide is the only source of sulfur in the flared gas and therefore determines the emission of sulfur dioxide. To assess this uncertainty, a sensitivity study with different hydrogen sulfide concentrations is given in Section 3.3.5.

## 3.3 Estimation of uncertainties

In the following section the most relevant uncertainties are presented, together with approaches for their assessment. This includes the uncertainty concerning the flare detection in the presence of cloud cover, the uncertainty in the determination of the emitted heat flow $H$ via the fraction of radiated heat $f$, the uncertainty in the choice of the IU14 parameters and the changes in flare operation from one year to another as well as the influence of the spatial variability of hydrogen sulfide in the natural gas on the sulfur dioxide emissions. Apart from Section 3.3.4 all uncertainty estimations are confined to TP15.

### 3.3.1 Uncertainty due to cloud cover

In Section 3.2 a climatological data set of flaring emissions ($E_{clim}$) was derived. When using this data set we are losing the day to day variation of the flaring emissions that is delivered by VNP. Although daily satellite observations are available, the problem arises that usually parts of the scene observed by the satellite are covered by clouds. In the following we will describe a method of how to derive





daily flaring emissions based on the climatological emissions ($E_{clim}$), a threshold of cloud coverage
($N_{th}$), and the actual detected flares at a certain day. This is illustrated schematically by Fig. 4.
The closed grey pie in the lower layer of Fig. 4A gives the climatological number of flaring boxes in
the research domain. At a certain day only within the green flaring boxes active flares are detected
by VNP. The flaring boxes that are indicated in grey are those at which no active flares were detected
by VNP, either because they are inactive or obscured by clouds. We now further separate this grey
area by introducing an empirical threshold value $N_{th}$ of cloud cover. In areas that belong to the grey
fraction in Fig 4A, where the cloud cover is above $N_{th}$, we assume that the flares boxes are active and
emit with their climatological emission values (since there are no current observations available).
Those flare boxes are indicated by the dark blue color in Fig 4B. The light blue area indicates flare
boxes where the cloud cover is below $N_{th}$ and where no flares are detected by VNP. For this area we
postulate that all flare boxes are inactive and consequently have zero emissions. Finally we calculate
the total emissions at a certain day for $N_{th}$=50% ($E_{50}$), 75% ($E_{75}$) and 90% ($E_{90}$) as the sum of the
climatological emissions in the dark blue area and the directly detected flares in the green area.

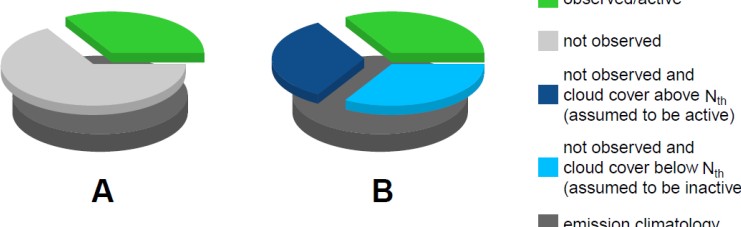

**Fig.4.** Pie charts illustrating the flaring emission uncertainty assessment due to cloud cover for TP15. The entirety of the
flare boxes within the emission climatology ($E_{clim}$) is given as closed grey pie in the bottom of **A** and **B**. **A** distinguishes
between flare boxes in which flares are detected at a certain day (green) and the complement of undetected flare boxes
(light grey). In **B** the light grey slice of **A** is separated in a cloud-covered (above cloud cover threshold $N_{th}$, dark blue) and
cloud-free (below $N_{th}$, light blue) by using remote sensing observations. Flare boxes which are not detected by VNP and
simultaneously show a cloud cover above $N_{th}$, are taken as active. Flare boxes which are not detected by VNP and
simultaneously show a cloud cover below $N_{th}$, are taken as inactive. For $N_{th}$ the values 50%, 75% and 90% are used. The
higher $N_{th}$ the smaller the dark blue slice in **B**.
To separate the light grey slice in Fig. 4A in covered and uncovered flare boxes, we used
instantaneous cloud fractional cover (CFC) from the geostationary Meteosat Second Generation 3
(MSG3) (CM SAF, 2015, copyright (2015) EUMETSAT) for every day of TP15 around the time of VNP
observation (Suomi-NPP overflight approx. at 1 UTC). This method is applied to all days of TP15 for
every flare box.
To ensure a consistent timing between cloud observation and VNP observation, the spatial domain
was reduced with a focus on the Niger Delta area (see Fig. 5a) and the flares were allocated
according to the cloud data grid with a mesh size of 0.03°.



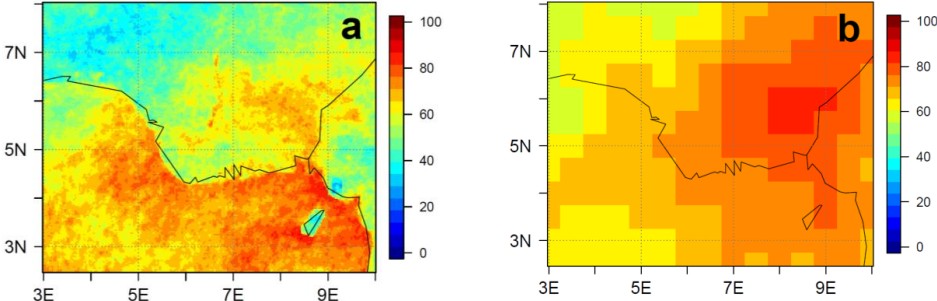


**Fig.5.** Fractional cloud cover (%) observed from (a) the geostationary MSG3 and (b) the sun-synchronous Aqua/AIRS,
averaged over TP15 around the time of VNP observation (approx. 1 UTC).

Fig. 5a shows that the onshore flaring area for TP15 is in mean covered with clouds by 50-70%. For
the offshore flaring area it is even higher with 70-90%. Therefore it is very likely that flares are
frequently masked by clouds and therefore not detected by VNP. However, we suspect that the
MSG3 cloud product underestimates (overestimates) the onshore (offshore) cloud cover when
comparing with the findings of van der Linden et al. (2015). The high offshore coverage and the
distinct land-water separation might be caused by overestimating low clouds in the presence of a
warm and moist tropical ocean.
Fig. 5b shows a cloud climatology using Aqua/AIRS Nighttime data (Mirador, 2016). The Aqua/AIRS
climatology shows higher cloud cover over land and no distinct separation between water and land
surface. Both products identify the highest onshore cloud cover in the northeast of Port Harcourt
(4.8°N, 7.0°E) and have similar values in the Nigerian offshore region (containing the offshore flares)
of about 70-80%. The major difference in the climatologies appears onshore between 4.5°N and 6°N.
This area includes the majority of the Nigerian onshore flares. Although it is not the aim of this study
to identify the most reliable cloud climatology for SWA, it has to be considered that MSG3 likely
underestimates the mean cloud cover over the Nigerian onshore flares up to 30%.
However, in the following the cloud climatology derived from MSG3 (Fig. 5a) is used since Aqua/AIRS
cannot provide the full spatial coverage for every day (due to the sun-synchronous orbit of
Aqua/AIRS).

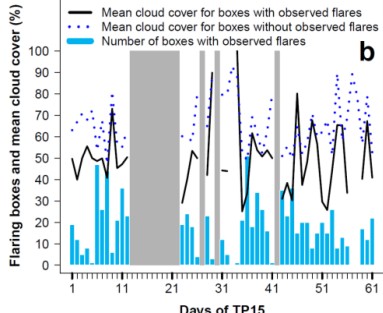

**Fig.6.** Number of boxes with detected flares per day (blue bars) and the mean fractional cloud cover for the boxes with
(without) detected flares as black solid (blue dotted) line (using MSG3, compare Fig. 5a). For the calculation of the latter,
the cloud cover of the non-active flare boxes within $E_{clim}$ are averaged (compare Niger Delta area in Fig. 2b). The grey
shaded areas are omitted due to lack of VNP observation.





Fig. 6 shows the number of grid boxes with active flares per day in TP15 as blue bars. The grey areas
indicate data gaps in VNP. $E_{clim}$ includes 185 flare boxes according to the domain in Fig 5a. For TP15
not more than 51 flare boxes are detected at once. In average only 8% of the total flaring area is
active at once. As expected the temporal evolution of the flare boxes and the cloud cover for these
boxes (black solid line in Fig. 6) shows an anticorrelation. The highest number of flare boxes at day 36
is reached in a period of a comparatively low cloud cover. The mean cloud cover for the non-active
flare boxes of $E_{clim}$ (blue dotted line in Fig. 6), is in general higher than for the active flare boxes which
implies that the cloud cover reduces the VNP detections. Fig. 6 also reveals that it is not suitable to
use the strict cloud-free condition for the separation in Fig. 4B because nearly all of the boxes would
be assigned to the dark blue cloud covered fraction and the resulting emissions would be nearly the
same as $E_{clim}$.
However, it has to be considered that the light points of flares are extremely small-scale signals
(1/5000 of the VNP pixel, Zhang et al. (2015)) and even for an almost completely closed cloud deck
VNP detections are possible.
The climatology $E_{clim}$ is the reference for this study. In addition we define $E_{obs}$ which only considers
the actually observed flares per day. $E_{50}$ is defined as the combination of actually observed flares and
cloud covered flares (see Fig. 4) with a cloud cover threshold of 50%. $E_{75}$ ($E_{90}$) is equal to $E_{50}$ but uses
a cloud cover threshold of 75% (90%).
To emphasize the difference between the different emission estimates, Fig. 7 shows the daily
emissions of $CO_2$ for TP15 as a spatial sum over the Niger Delta area (see Fig. 4a). In contrast to $E_{clim}$
(black dashed line), $E_{obs}$, $E_{50}$, $E_{75}$ and $E_{90}$ (solid lines) have a temporal variation within TP15.

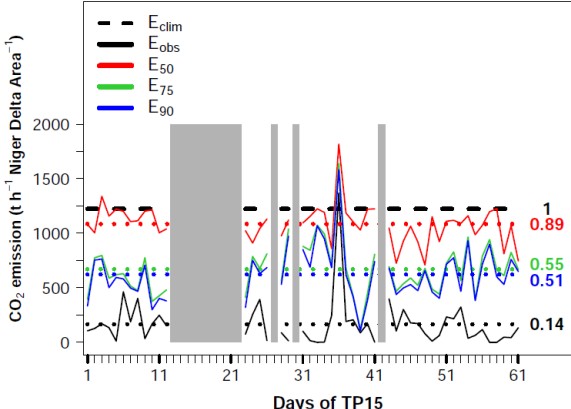

**Fig.7.** Daily $CO_2$ emissions (kg h$^{-1}$) within TP15 from flaring summed up over the Niger Delta area defined in Fig. 4a for the five emission estimates: $E_{clim}$ (climatology, black dashed line), Eobs (VNP observations, black solid line), $E_{50}$ (combination of VNP observations and the climatology for a cloud cover threshold of 50%, red solid line), $E_{75}$ (as $E_{50}$ but for a cloud cover threshold of 75%, green solid line), $E_{90}$ (as $E_{50}$ but for a cloud cover threshold of 90%, blue solid line). The dotted lines denote the spatiotemporal average of $E_{obs}$, $E_{50}$, $E_{75}$ and $E_{90}$. The numbers on the right hand side show the ratios of the spatiotemporal averages $E_{obs}$, $E_{50}$, $E_{75}$ and $E_{90}$ towards $E_{clim}$. The grey shaded areas are omitted due to lack of VNP observation.


$E_{clim}$ delivers a daily $CO_2$ emission of about 1250 t h$^{-1}$ within the Niger Delta area. The pure daily VNP
observations within $E_{obs}$ (black solid line) show only 14% of $E_{clim}$ emissions (numbers on left hand side
of Fig. 7) on average (black dotted line). The emissions from VNP observations together with the
climatology for the cloud threshold of 50% within $E_{50}$ (red solid line) is closest to the climatology (89%



of $E_{clim}$, red dotted line). The high overall cloud cover within the domain (compare with blue dotted
line in Fig. 6) together with the relative low cloud cover threshold leads to the result, that nearly the
complete climatology is used for $E_{50}$ and therefore the difference to $E_{clim}$ is small. The emissions from
VNP observations together with the climatology for the cloud threshold of 75% and 90% within $E_{75}$
and $E_{90}$ (green and blue solid line) shows only small deviations but are significantly reduced in
comparison to $E_{clim}$ (55% and 51% of $E_{clim}$, green and blue dotted line). Day 36 of TP 15 shows highest
emissions in $E_{obs}$, $E_{50}$, $E_{75}$ and $E_{90}$, owing to the combination of low cloud cover and high flaring
activity (compare with Fig. 6). Regarding the uncertainty in the cloud cover climatology (compare Fig.
5a and Fig. 5b), the emissions of $E_{50}$, $E_{75}$ and $E_{90}$ might be underestimated. The underestimation of
the cloud cover in the onshore flaring area could lead to an unjustified increase in flare boxes below
$N_{th}$ and therefore to a reduced number of active flares per day.
These emission estimations contain different information. $E_{clim}$ includes all flares of the domain
despite cloud cover but can overestimate the emissions. $E_{obs}$ shows the VNP reality, including a
temporal development, but cannot consider the cloud-covered flares. $E_{50}$, $E_{75}$ and $E_{90}$ combine the
flare location information of $E_{clim}$ and the full temporal resolution of VNP in $E_{obs}$ by using cloud
observations. However this approach is based on the assumption that all cloud covered flare boxes
are active, which is also linked to high uncertainty. Additionally $E_{50}$, $E_{75}$ and $E_{90}$ depend on the
availability of a longer VNP observational dataset. The ratios of the spatiotemporal means of $E_{obs}$, $E_{50}$,
$E_{75}$ and $E_{90}$ to the spatial mean of $E_{clim}$ (as denoted by the numbers in Fig. 7) are used as correction
factors (CF) for $E_{clim}$ in the following (see Tab. 4). $E_{clim}$ is taken as the reference (CF=1).

**Tab.4.** Emission estimations including information about flaring (daily observation and climatology) and cloud cover
observation. The correction factors (CF) are derived for TP15 from a spatiotemporal emission mean in the Niger Delta area
(2.5°N-8°N, 3°E-10°E) and refer to $E_{clim}$.

| Name | Emission estimate | CF for $E_{clim}$ |
|---|---|---|
| $E_{clim}$ | Climatology (reference) | 1 |
| $E_{obs}$ | Observed flares | 0.14 |
| $E_{50}$ | Observed flares + climatology ($N_{th}$= 50%) | 0.89 |
| $E_{75}$ | Observed flares + climatology ($N_{th}$ = 75%) | 0.55 |
| $E_{90}$ | Observed flares + climatology ($N_{th}$ = 90%) | 0.51 |


These CF are a simple method to include the information of $E_{obs}$, $E_{50}$, $E_{75}$ and $E_{90}$ into $E_{clim}$ by
multiplying $E_{clim}$ with the corresponding correction factor. In this case the same 185 flare boxes of
$E_{clim}$ are used but with an emission strength reduced to the averaged conditions of $E_{obs}$, $E_{50}$, $E_{75}$ and
$E_{90}$. This approach is based on the assumption that the correction factor, deduced for the Niger Delta
area, is valid for the whole domain specified in Section 3.1. This assumption seems to be justified
since the Niger Delta area contains most of the gas flares in the domain.

*3.3.2 Uncertainty due to IU14 input parameters*

To assess the uncertainty which arises from the combustion efficiency $\eta$ and the availability of
combustion air $\delta$, a sensitivity study has been carried out. The exact values for the SWA flares are
unknown and very likely highly variable from one flare to another, depending on the flare type and
operation. Fig. 8a shows the flare emissions averaged over SWA and TP15 for CO, $CO_2$, NO and $NO_2$.
The parameters $\eta$ and $\delta$ are varied referring to IU14. A complete combustion ($\eta = 1$) does not
produce CO emissions since all carbon is transformed to $CO_2$ (not shown). With decreasing $\eta$ and $\delta$,
the CO and $CO_2$ emissions increase. Concerning CO we assume the lower limit for $\eta = 0.9$ and



$\delta = 1.3$ (left of Fig. 8a) and the upper limit for $\eta = 0.5$ and $\delta = 0.76$ (right of Fig. 8a). The values
used for this study are located in the center of Fig. 8a. By taking the latter as reference, the lower
(upper) limit leads to a decrease (increase) in CO emission of -63% (+210%). For $CO_2$ we derived an
upper (lower) limit of +38% (-72%).
A higher combustion efficiency or a higher availability of combustion air allows an enhanced
formation of NO. Therefore NO emissions increase (decrease) with decreasing $\eta$ ($\delta$). We assume the
lower limit for $\eta = 0.9$ and $\delta = 0.95$ and the upper limit for $\eta = 0.5$ and $\delta = 1.30$. Taking again the
central parameter set of Fig. 8a as reference, the lower (upper) limit leads to a decrease (increase) in
NO emission of -77% (+441%).

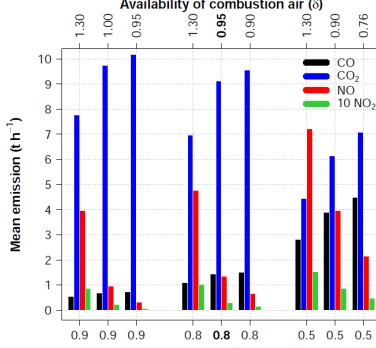
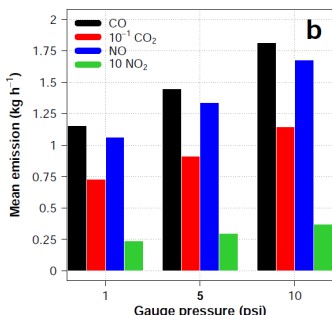


**Fig.8.** Flaring emissions (kg h$^{-1}$) spatiotemporally averaged over SWA and TP15 depending on (a) combustion efficiency $\eta$
and availability of combustion air $\delta$ and (b) gauge pressure (psi) for the setup of $\eta$ and $\delta$ which is used for this study
(emphasized in bold). $SO_2$ is not shown because it does not depend on $\eta$ or $\delta$.

The emissions of $NO_2$ are comparatively low owing to the source temperature which is in general
lower than the $NO_2$ formation threshold of 1600 K.
In addition, Fig. 8b shows the emissions depending on the gauge pressure for 1 (lower limit), 5 and
10 psi (upper limit) (7, 34 and 69 kPa respectively) for $\eta = 0.8$ and $\delta = 0.95$. Regarding 5 psi as the
reference, the lower (upper) limit leads to a decrease (increase) in CO emissions of -21% (+26%).
Fig. 8 emphasizes that the technical conditions of flaring crucially influence the emission strength and
that the emissions are more sensitive towards $\eta$ and $\delta$ than towards the gauge pressure.

*3.3.3 Uncertainty due to the fraction of radiated heat*

To estimate the uncertainty in the fraction of radiated heat $f$ (see Tab. 2), we have used the standard
deviation of the literature values given in the appendix of Guigard et al. (2000) in addition to the
mean value of $f = 0.27$. This leads to a domain of uncertainty for the value $f$ of ($^{0.38}/_{0.16}$). Therefore
the VNP observed radiant heat is multiplied with the factor $1/f$ of 3.7 ($^{6.2}/_{2.6}$).

*3.3.4 Interannual variability*

The differences in flaring between TP14 and TP15, indicated in Fig. 1 and Fig. 2, are quantified in this
section according to the emissions of CO (Fig. 9a) and $CO_2$ (Fig. 9b). The boxplots include all flaring
boxes for the two domains SWA (green) and the Niger Delta area (blue). The numbers above indicate
the integrated emissions per hour and area in tons.




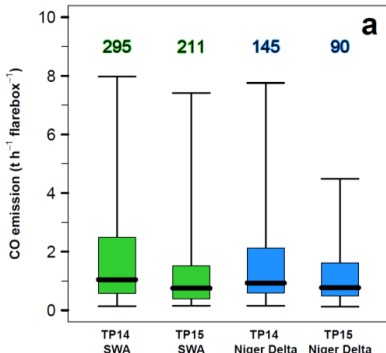 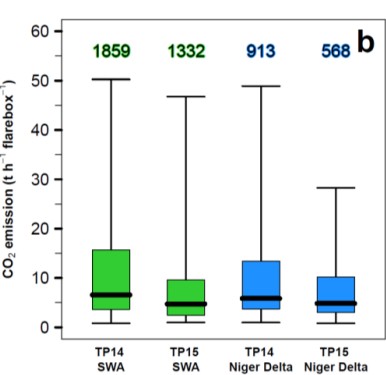


**Fig.9.** Flare box emissions of (a) CO and (b) $CO_2$ ($E_{clim}$, t h$^{-1}$ flarebox$^{-1}$) for SWA (green) and the Niger Delta area (blue) for
TP14 and TP15. The values above the boxplots indicate the emissions per hour, integrated over SWA (green) and the Niger
Delta area (blue). The whiskers span the data range from the 0.025-quantile to the 0.975-quantile (95% of the data). Data
outside of this range is not shown.

The emissions of $CO_2$ are 6.3 times higher than the CO emissions. For SWA the mean value of
emissions is statistically significant lower for TP15 compared to TP14 (Wilcoxon-Mann-Whitney rank
sum test with a significance level of 0.05). For the Niger Delta area the emission means show no
significant difference. The significant different mean values for SWA emissions emphasize the
relevance of using a flaring dataset which is up to date to reduce uncertainties arising from
deviations in flare locations or flaring processes.

*3.3.5 Uncertainty due to spatial variability in $H_2S$*

Since hydrogen sulfide ($H_2S$) is the only sulfur source in the flared gas, it determines the emission of
sulfur dioxide. The natural gas composition measurements from the ten flow stations given in
Sonibare and Akeredolu (2004) contain only two stations with nonzero $H_2S$ content. Therefore
averaging over the ten stations (see Tab. 3) leads to a low $H_2S$ content in the emission calculations.
By using the highest concentration value of $H_2S$ given in Sonibare and Akeredolu (2004) (see Tab. 3,
$H_2S$ concentration 0.03% instead of 0.005%), we try to estimate the upper limit of $SO_2$ emission,
assuming that all flares are provided with this more sulfur containing gas. With this approach the
spatiotemporal averaged $SO_2$ emissions increase from 0.6 to 4.9 kg h$^{-1}$. The maximum values in the
flare boxes increase from 4.7 to 41.8 kg h$^{-1}$. These are rather low values.
This comparison reveals that among the flaring conditions also the natural gas composition plays an
important role in estimating the flaring emissions reasonably. To rely on a single measurement
dataset for a large flaring domain and without taking into account spatial variability is therefore
problematic but has to be accepted owing to insufficient data.
This section has estimated the uncertainties in gas flaring due to cloud cover, parameters of IU14, the
fraction of radiated heat, the temporal variability and the $H_2S$ concentration in the natural gas. The
uncertainty regarding the spatial variability of the total hydrocarbon fraction of the natural gas,
which is estimated by the variations in the ten flow station measurements of Sonibare and Akeredolu
(2004), is below 1%.
However, there are further assumptions or sources of uncertainty which cannot be quantified within
this study: We assume that the natural gas composition, which is measured in one region, is valid for





SWA entirely. The gas flares are taken as constant emission sources because VNP only provides one
observation (overflight) per day. We cannot take into account the spatial variability of the flares
concerning the IU14 parameters and the stack heights. And finally IU14 delivers no VOCs and black
carbon.

## 4. Comparison with existing emission inventories

The following section places the estimated flaring emissions of this study in the context of existing
emission inventories, by taking the focus on $CO_2$. A direct comparison with existing emission
inventories is problematic due to different reference time periods, spatial domains, definitions of
emission sectors and the limitation of chemical compounds. Tab. 5 summarizes the $CO_2$ emissions for
different inventories regarding Nigeria or the Niger Delta area as denoted in Fig. 5a, the flaring
hotspot of the research domain. The results of this study shows no flaring in the northern part of
Nigeria and therefore flaring within the Niger Delta area can be seen as the total flaring area of the
country. To derive annual emission values for the results of this study, it is assumed that the flaring
emission conditions of TP14 and TP15 are representative for the whole year 2014 and 2015
respectively. Therefore the hourly emissions are integrated over 365 days.

**Tab.5.** Comparison between existing emission inventories for $CO_2$ (with a focus on gas flaring if available) and the results of this study for Nigeria or the Niger Delta area in teragram (Tg) per year. For TP14 and TP15 it is assumed that the two month observations represent the flaring conditions of the whole year 2014 and 2015 respectively. Therefore the emissions were integrated to yearly values. The values in brackets represent the upper and lower limit owing to the uncertainties estimated in Section 3. For the fraction of radiated $f$ the mean value 0.27 and the lower (upper) boundary of 0.16 (0.38) are used, representing a further source of uncertainty. The products given in bold are directly related to flaring emissions.

| Emission inventory | Time period | $CO_2$ emissions (Tg y$^{-1}$) | | |
| --- | --- | --- | --- | --- |
| | | $f = 0.16$ | $f = 0.27$ | $f = 0.38$ |
| **This study (E$_{clim}$)** | 2014 (from TP14) | 13 ($^{19}/_3$) | 8 ($^{12}/_2$) | 6 ($^8/_1$) |
| **This study (E$_{clim}$)** | 2015 (from TP15) | 8 ($^{12}/_2$) | 5 ($^7/_1$) | 4 ($^5/_0$) |
| **This study (E$_{75}$)** | 2014 (from TP14) | 7 ($^{11}/_2$) | 4 ($^7/_1$) | 3 ($^5/_0$) |
| **This study (E$_{75}$)** | 2015 (from TP15) | 5 ($^7/_1$) | 3 ($^4/_0$) | 2 ($^3/_0$) |
| **CDIAC (2015b)**[1] | 2011 | | 27.47 | |
| **EIA (2015)**[2] | 2010; 2011; 2013 | | 38.81; 41.39; 52.83 | |
| **Doumbia et al. (2014)**[1] | 2010 | | 45 | |
| **EDGAR 4.2**[3] (ECCAD, 2015) | 2008 | | 8.75 | |
| **EDGAR 4.2**[4] (ECCAD, 2015) | 2008 | | 3.50 | |
| EDGAR 4.3.2[5] (EDGAR, 2016) | 2010; 2011; 2012 | | 29.4, 28.8, 28.9 | |
| EDGARv43FT2012[6] (EDGAR, 2014) | 2014 | | 93.87 | |

[1] from gas flaring, Nigeria
[2] from consumption and flaring of natural gas
[3] from refineries and transformation, Nigeria
[4] from refineries and transformation, Niger Delta area according to Fig. 5a
[5] from venting and flaring of oil and gas production, Nigeria
[6] emission totals of fossil fuel use and industrial processes (cement production, carbonate use of limestone and dolomite, non-energy use of fuels and other combustion). Excluded are: short-cycle biomass burning (such as agricultural waste burning) and large-scale biomass burning (such as forest fires), Nigeria

The $CO_2$ emission estimations of this study are given in Tab.5 together with an overall uncertainty
range ($^{+38}/_{-72}$ %) including the uncertainty from the IU14 parameters $\eta$ and $\delta$ ($^{+12}/_{-52}$ %) and the
gauge pressure ($^{+26}/_{-21}$ %) and from spatial variability of total hydrocarbon. The latter uncertainty is
small (below 1%) owing to the low variation in THC concentration in the measurements of Sonibare



559 and Akeredolu (2004). The uncertainty due to cloud cover is represented by $E_{75}$. Regarding the

560 relatively large uncertainty there is no preference in one of the emission estimates $E_{clim}$ and $E_{obs}$.

561 By assuming the uncertainty range of the fraction of radiated heat $f$ between 0.16 and 0.38, the

562 results of the study on hand show $CO_2$ emissions in the same order of magnitude as the Carbon

563 Dioxide Information Analysis Center (CDIAC, 2015b), the Energy Information Administration (EIA,

564 2015) and the EDGARv.4.3.2 (EDGAR, 2016) database, with best results for $f = 0.16$ but with an

565 overall tendency to underestimate the emissions. $E_{clim}$ shows smaller deviations to the existing

566 inventories than the cloud correction approach of $E_{75}$. A direct comparison is hindered by a time lag

567 of 3-4 years and missing information about the uncertainties of CDIAC. The values of EIA are higher

568 than those of CDIAC because EIA includes the consumption of natural gas in addition to gas flaring.

569 Doumbia et al. (2014) combines Defense Meteorological Satellite Program (DMSP) observations of

570 flaring with the emission factor method to derive flaring emissions. The results agree with EIA (2015)

571 but are 64% higher than CDIAC (2015b).

572 The emission inventory EDGAR v4.2 (ECCAD, 2015) delivers 8.75 (3.50) Tg $CO_2$ $y^{-1}$ for Nigeria (Niger

573 Delta area) for the emission sector *refineries and transformation*, which is in good agreement with

574 the results for the study on hand.

575 As a benchmark for the flaring $CO_2$, the total $CO_2$ emissions for Nigeria are given by EDGAR (2014),

576 (fossil fuel use and industrial processes). Taking EDGAR (2014) as a reference for total $CO_2$ emissions

577 of Nigeria, flaring emissions contributes by 9 $\binom{13}{2}$% (2014; $E_{clim}$ $f = 0.27$), 14 $\binom{20}{3}$% (2014; $E_{clim}$

578 $f = 0.16$), 9% (2008; ECCAD, 2015), 28% (2011; CDIAC, 2015b), 48% (2010; Doumbia et al., 2014) or

579 56% (2013; EIA, 2015). The large spread between the different inventories emphasizes the large

580 uncertainty within the estimation of emissions from gas flaring.

581 A shortcoming of the PEGASOS_PBL-v2 (not shown) and the EDGAR v4.2 emission inventory is the

582 lack of offshore flaring emissions in the Gulf of Guinea south of Nigeria. For CDIAC and EIA this

583 cannot be verified since the data is only available as a single value per country.

584 The differences between the results of this study and the existing emission inventories might be

585 caused by an underestimation of the flow rate by VNP and Eq. 1 or by an inconsistent definition of

586 emission source sectors for the existing inventories. $E_{clim}$, $E_{75}$, Doumbia et al. (2014) and CDIAC

587 (2015b) focus on gas flaring, whereas other products also include natural gas consumption and

588 emissions from refineries and transformation which also can include non-flaring emissions within and

589 outside the areas indicated as flaring area by the satellite imagery. In addition, the existing

590 inventories do not provide current values (time lag of 2 to 6 years) and therefore not consider the

591 emission reduction indicated by Fig. 9.

592

593 **5. Discussion and conclusions**
594

595 The gas flaring emission estimating method of Ismail and Umukoro (2014) (IU14) has been combined

596 with the remote sensing flare location determination of the VIIRS Nightfire Prerun V2.1 Flares only

597 (VNP) (VIIRS, 2015) for a new flaring emission parameterization. The parameterization combines

598 equations of incomplete combustion with the gas flow rate derived from remote sensing parameters

599 instead of using emission factors and delivers emissions of the chemical compounds CO, $CO_2$, NO and

600 $NO_2$.

601 Within this study the parameterization was applied to southern West Africa (SWA) including Nigeria

602 as the second biggest flaring country. Two two-month flaring observation datasets for June-July 2014

603 and 2015 were used to create a flaring climatology for both time periods. In this climatology all

604 detected flares emit with their mean activity.



The uncertainties owing to missed flare observations by cloud cover, parameterization parameters, interannual variability and the natural gas compositions were assessed. It can be shown that the highest uncertainties arise from the definition of the fraction of radiated heat $f$ and the IU14 parameters. By using remote sensing cloud cover observations, a correction factor for the flaring climatological emission was derived which reduces the mean emissions about 50%. However, owing to the large uncertainty ranges, no significant difference between the climatological inventory and the cloud corrected inventory can be stated. Comparing the emissions of 2014 and 2015, a reduction in the flaring area, density of active flares and a significant reduction in SWA emissions about 30% can be observed, which underlines the need for more recent emission inventories.

The uncertainty due to the natural gas composition is compound dependent. The spatial variation in total hydrocarbon is negligible but the availability of hydrogen sulfide, which exclusively determines the amount of emitted $SO_2$, cause large uncertainty By taking the combustion efficiency to derive the fraction of unburned natural gas, the amount of emitted VOCs might be estimated in addition to the species of the study on hand but would also be linked to high uncertainties concerning the VOC speciation. The uncertainty in VOC emission is increased drastically by natural gas which is vented directly into the atmosphere instead of being flared, since the venting cannot be detected by VNP.

With a focus on Nigeria, the $CO_2$ emission estimates of this study were compared with existing inventories. For the climatology, $CO_2$ emissions of 8 $\binom{12}{2}$ Tg y$^{-1}$ for 2014 and 5 $\binom{7}{1}$ Tg y$^{-1}$ for 2015 were derived. EDGAR v4.2 for the year 2008 shows the same order of magnitude when limiting to emissions from refineries and transformation. CDIAC (Carbon Dioxide Information Analysis Center) is in the same order of magnitude as the results of this study. Doumbia et al. (2014) and EIA (Energy Information Administration) show emissions which are 2.4 and 2.8 times higher than the results of this study. The deviations might be caused by uncertainties in the flow rate derived by VNP radiant heat, which can be assessed only rudimentary via the parameter of the fraction of radiated heat. Additionally, the usage of emission factors in the existing inventories which did not take into account the spatiotemporal variability of flaring, inconsistent emission sector definitions or the time lag of the emission inventories of 2-5 years can lead to deviations. The positive trend in Nigerian gas flaring $CO_2$ emissions derived by EIA from 38.81 to 52.83 Tg y$^{-1}$ between 2010 and 2013 contradicts the findings of Doumbia et al. (2014) and this study, which generally show a decrease in emissions from 1994 to 2010 and from 2014 to 2015, respectively. Based on the sensitivity study, which reveals high uncertainties of the flaring emission, we conclude that there is no preference in the choice of one of the emission estimates presented in this study. Therefore we recommend the use of the climatological approach when using the R package.

Despite the generally large uncertainties in the estimation of emissions from gas flaring, this method allows a flexible creation of flaring emission datasets for various applications (e.g. as emission inventory for atmospheric models). It combines observations with physical based background concerning the combustion. The use of current data makes it possible to consider present trends in gas flaring. Even the creation of near real-time datasets with a time lag of one day is possible. The emissions are merged on grid predefined by the user and depending on the availability of VNP data, the temporal resolution can be selected from single days to years.

An improvement of this parameterization can be achieved by an extension of the IU14 method to black carbon and VOCs and an inclusion of spatial resolved measurements of the natural gas composition in combination with information of the gas flaring processes from the oil producing industry.



## Acknowledgments

The research leading to these results has received funding from the European Union 7th Framework Programme (FP7/2007-2013) under Grant Agreement no. 603502 (EU project DACCIWA: Dynamics-aerosol-chemistry-cloud interactions in West Africa).

We are grateful to Godsgift Ezaina Umukoro (Department of Mechanical Engineering, University of Ibadan, Nigeria) for the kind support during the implementation of their combustion reaction theory into our parameterization. We also thank the Earth Observation Group (EOG) of NOAA for providing the *VIIRS Nightfire Flares Only product*.

## Code and/or data availability

This publication includes a package of well documented R scripts which is free available for research purposes and enables the reader to create their own gas flaring emission datasets. It includes exemplarily the preprocessing for June-July 2015 with a focus on southern West Africa. You get access to the code via zenodo.org (DOI: 10.5281/zenodo.50938).

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
