# Peer review of "Development of a new gas flaring emission data set for southern West Africa"

_Geoscientific Model Development, 2016_

## Referee Comment (RC1) · C. Elvidge (Referee) · 3 Aug 2016

C. Elvidge (Referee)

chris.elvidge@noaa.gov

There are several issues that would require some rework prior to publication:

1) The VIIRS Night Fire (VNF) "flares only" dataset is not suitable for scientific applications. It is generated by stripping out VNF detections with either no temperature or temperatures unser 1400K. This eliminates most biomass burning and ambiguous detections. The purpose of this is to provide a quick daily overview of global gas flaring activity. There are many times when a flare was detected in a single spectral band (usually M10 at 1.6 um), in which case the Planck curve cannot be fit and a temperature cannot be calculated. These detections have been lost in the dataset used by the authors. In addition, some flares are known to fluctuate in temperature and dip below 1400 K. These low temperature flaring events are also lost in the "flares only"

daily summaries. The produce a more thorough analysis, the authors should work from the original daily VNF files. At best the "flares-only" version of the data provides a 'quick-and-dirty" depiction of global gas flaring.

2) The authors do not account for variations in cloud cover. This can be done based on the VIIRS Cloud product suite.

3) The text should reference the following paper: Methods for Global Survey of Natural Gas Flaring from Visible Infrared Imaging Radiometer Suite Data (http://www.mdpi.com/1996-1073/9/1/14).

4) NOAA has global flaring data spanning 2012-2014 available at: http://ngdc.noaa.gov/eog/viirs/download_global_flare.html. There is a csv that contains locations and annual summaries of temperatures and radiant heat output of individual flares, normalized for cloud cover. The flared gas volume estimates are derived from an empirical calibration with CEDIGAZ reported flaring. It would be interesting to compare the NOAA results with those from the methods described in this paper.

5) In the last sentence of the first paragraph, the text references the World Bank for a set of national flared gas volume estimates. The text should make it clear that these estimates were produced by NOAA using DMSP satellite data. There is a new set of estimates derived from VIIRS data at http://ngdc.noaa.gov/eog/viirs/download_global_flare.html.

---

## Referee Comment (RC2) · C. Elvidge (Referee) · 5 Aug 2016

1. Being fully familiar with the flares only version of the VIIR Nightfire product I can certify that this product is not suitable for use in a scientific study. If the authors had contacted my team at the start of their study we could have explained this to them and directed them to the full VNF data files, which are suitable for use in scientific studies.

2. NOAA does provide cloud state for each VNF detection - from the VIIRS cloud product suite. There are four states: confidently cloudy, probably cloudy, probably clear, confidently clear. But what is not recorded in the VNF files are the number of clear observations where the flare was not detected. The NOAA annual gas flaring data takes this into account. I dispute the authors contention that "it is not necessary to account for the variations in cloud cover."

[Figure]

3. My overall impression is that these authors are willing to use data that are known to be flawed and ignore the effects of cloud cover variations in order to get a paper published without doing any addition work. If this journal is willing to publish papers with flaws like this disclosed - heaven help them.

———————————————————

---

## Author Comment (AC2) · 8 Aug 2016

Dear Dr. Elvidge,

thank you for your reply from 5 August 2016.

The authors are puzzled about the different tenor in the two reviews we have achieved from you. We regret that you have the impression we are not willing to invest additional work for this study. Under point 4 of our reply from 3 August 2016 we followed your idea to compare your dataset with our study. Maybe one of your team can give us information on how to use the full VNF data files correctly and how to separate the flaring sources from other combustion sources (e.g. forest fires). With this data set, under consideration of the cloud correction, we will repeat our study.

[Figure]

Sincerely yours,

Konrad Deetz and Bernhard Vogel

---

## Referee Comment (RC3) · C. Elvidge (Referee) · 9 Aug 2016

If the authors will send me an email listing the countries of interest and the time period, I can have temporal profiles extracted for each of the flaring sites we have identified.

---

## Author Comment (AC3) · 31 Aug 2016

Dear Dr. Elvidge,

thank you for your reply from 9 August 2016. As proposed we have send you the list of countries and the time period of interest. We have received the temporal profiles of the flaring sites from your working group. Thanks to Dr. Mikhail Zhizhin for the data processing and the kind support. We have completely revised the manuscript and the source code according to the new data. Thank you for calling our attention to the limitations of the "VIIRS Flares Only" product.

Kind regards

Konrad Deetz and Bernhard Vogel

---

## Referee Comment (RC4) · M. Zhizhin (Referee) · 14 Oct 2016

In the paper a new method to model emissions from gas flaring is developed and validated on oil fields in Western Africa. The paper is a substantial contribution to the modeling science, and the approach is valid and motivating for further research.

I have some comments on the presentation and details of the method which could be considered by the Authors before it is published.

0. I would recommend changing abbreviation VNP (VIIRS Nightfire Product) to commonly used VNF (simply VIIRS Nightfire) in the manuscript.

1. Formula (1) derives gas flow rate from flare radiative heat and temperature measured from satellite. It is a basis of the proposed model. However, it is taken from

Appendix of regulating document by the German Environmental protection agency. This is technical, not scientific source. The derivation of the formula is not provided neither in the paper under review, nor in the cited document. The cited document has no source for the formula either. It is important to derive the formula (1) or to provide a scientific reference.

2. Flare temperature used in the formula (1) is taken from instantaneous satellite measurement (VNF). It has a large variance depending on atmospheric conditions etc. I would recommend using mean flare temperature averaged over all cloud-free detections.

3. The number 283 used in the formula (1) I believe stands for ambient air temperature at night? Is it a proper climatological value for Wester Africa ?

4. Comments 1-3 may result in a wider variance of the proposed model output, and the model sensitivity analysis should be presented.

5. The Authors have made a considerable effort to take into account cloud conditions which can mask flare observations from space. Why not to use only cloud-free observation days, and to count detected/not detected flare cases to derive mean radiative heat ?

I would like to acknowledge that the Authors provide software sources and input data used in the study as the paper supplement. It is helpful for reproduction and reuse of their science and model.

―――――――――――――――――

---

## Author Response (AR1)

Dear Dr. Elvidge (Referee, Geoscientific Model Development), thank you for your reviewer report from 3 August 2016. We have accounted for the comments and suggestions in the revised manuscript version. Please find our replies to the particular comments in the following.

Sincerely,

Konrad Deetz and Bernhard Vogel

Referee comments:

1) The VIIRS Night Fire (VNF) "flares only" dataset is not suitable for scientific applications. It is generated by stripping out VNF detections with either no temperature or temperatures under 1400K. This eliminates most biomass burning and ambiguous detections. The purpose of this is to provide a quick daily overview of global gas flaring activity. There are many times when a flare was detected in a single spectral band (usually M10 at 1.6 um), in which case the Planck curve cannot be fit and a temperature cannot be calculated. These detections have been lost in the dataset used by the authors. In addition, some flares are known to fluctuate in temperature and dip below 1400 K. These low temperature flaring events are also lost in the "flares only" daily summaries. The produce a more thorough analysis, the authors should work from the original daily VNF files. At best the "flares-only" version of the data provides a 'quick-and-dirty" depiction of global gas flaring.

For our work in the project Dynamics-aerosol-chemistry-cloud interactions in West Africa (DACCIWA) we wanted to have a consideration of gas flaring in our regional atmospheric model which includes the up-to-date characteristics of southern West Africa (SWA). The DACCIWA measurement campaign took place in June/July 2016 and for this time we need the flaring information for our model. Emission estimates for 2012/2013/2014 are not meaningful in our case, because the emissions are not constant from year to year. Also your new estimation (http://ngdc.noaa.gov/eog/viirs/download_global_flare.html) shows a decrease in flaring for Nigeria. To use older data would lead to overestimations.

The SWA emission inventory for flaring was not available when we started our research. The presented method is therefore our first approach to tackle the problem with the missing flaring emissions in our atmospheric chemistry simulations. Instead of using just constant emissions factors for flaring, we now have very regional information available.

We are concentrating on the description of the air pollution in our modelling system COSMO-ART and try to include all relevant emission sources. We are no experts in extracting the flaring sources from the general combustion sources detected by VIIRS Nightfire. Therefore we have relied on the "flares only" product published at http://ngdc.noaa.gov/eog/viirs/download_viirs_flares_only.html. Even if the data basis for our study is not perfect regarding VNF, there is a strong progress compared to the state before. We have changed our manuscript according to this problem. We have remarked, that the use of the "flares only" product is just a first approach and that this data contains greater uncertainties compared to the original VNF product. Future users of this parameterization can change the VNP input. The general method of the parameterization will not be affected by that.

2) The authors do not account for variations in cloud cover. This can be done based on the VIIRS Cloud product suite.

I see your point but our study focus is located to the creation of an emission dataset based on a VNF climatology rather than taking the VNF data day by day. In section 3.3.1 we describe the problem of flares that are masked by clouds (and the overall question whether the flare below the cloud is active or not) in detail and assess the uncertainty by using remote sensing cloud data from MSG and Aqua/AIRS. By deriving a flaring climatology (over two month), we are able to identify all flares (even if there are sometimes covered by clouds). With this climatological approach we get the mean emission strength of every flare (more precisely for every flare box). Therefore it is not necessary to account for the variations in cloud cover. Even if we would know, that a certain flare is masked by clouds at a certain day we don't know whether this flare is currently active and how large the radiant heat is. When we use our flaring climatology in our regional atmospheric model, all available flares are active at once with their mean emission strength.

3) The text should reference the following paper: Methods for Global Survey of Natural Gas Flaring from Visible Infrared Imaging Radiometer Suite Data (http://www.mdpi.com/1996-1073/9/1/14).
We agree on that and have referenced the publication.

4) NOAA has global flaring data spanning 2012-2014 available at: http://ngdc.noaa.gov/eog/viirs/download_global_flare.html. There is a csv that contains locations and annual summaries of temperatures and radiant heat output of individual flares, normalized for cloud cover. The flared gas volume estimates are derived from an empirical calibration with CEDIGAZ reported flaring. It would be interesting to compare the NOAA results with those from the methods described in this paper.

From the xlsx file VIIRS_Global_flaring_d.7_slope_0.331_web.xlsx we have selected the 193 available Nigerian upstream flares and selected the flares which have a detection frequency greater than zero for 2014. We assume that "Avg. K" mean source temperature in K and "Ellipticity" means the radiant heat in MW. This data we have used as input for the parameterization presented in this study (with the same configuration). Finally we have integrated the volume stream of all Nigerian flare boxes from m-3 s-1 to m-2 y-1 and finally transformed it to bcm. The result is 8.55 bcm (271.0391 m-3 s-1). In the xlsx file the flared volume is estimated as 8.442995283 bcm for Nigeria in 2014. So if we use the same source temperature and radiant heat input as Elvidge et al. (2015) for Nigeria in 2014, we can reproduce the estimated flared volume with our method with a deviation below 1.3%.

Within our VNP data set for 2014 we estimate the flaring to 29.8 bcm. Regarding the uncertainty range of this estimation, the value is approx. by a factor of two higher than the other inventory. The uncertainty might result from the uncertainties in the estimation of the gauge pressure $p_g$ and the fraction of the total reaction energy that is emitted as radiation f. In our flaring climatology we assume that all available flares are active at once with their mean emission strength, so this could lead to the higher values of the flared volume.

5) In the last sentence of the first paragraph, the text references the World Bank for a set of national flared gas volume estimates. The text should make it clear that these estimates were produced by NOAA using DMSP satellite data. There is a new set of estimates derived from VIIRS data at http://ngdc.noaa.gov/eog/viirs/download_global_flare.html.

We agree on that and have changed the manuscript accordingly. A remark to the availability of updated global flaring estimates for 2013 and 2014 at http://ngdc.noaa.gov/eog/viirs/download_global_flare.html are mentioned in the manuscript.

Dear Dr. Elvidge (Referee, Geoscientific Model Development), thank you for your reviewer report from 5 August 2016. We have accounted for the comments and suggestions in the revised manuscript version. Please find our replies to the particular comments in the following.

Sincerely,

Konrad Deetz and Bernhard Vogel

Referee comments:

1.) Being fully familiar with the flares only version of the VIIR Nightfire product I can certify that this product is not suitable for use in a scientific study. If the authors had contacted my team at the start of their study we could have explained this to them and directed them to the full VNF data files, which are suitable for use in scientific studies.

2.) NOAA does provide cloud state for each VNF detection - from the VIIRS cloud product suite. There are four states: confidently cloudy, probably cloudy, probably clear, confidently clear. But what is not recorded in the VNF files are the number of clear observations where the flare was not detected. The NOAA annual gas flaring data takes this into account. I dispute the authors contention that "it is not necessary to account for the variations in cloud cover."

3.) My overall impression is that these authors are willing to use data that are known to be flawed and ignore the effects of cloud cover variations in order to get a paper published without doing any addition work. If this journal is willing to publish papers with flaws like this disclosed - heaven help them.

The authors are puzzled about the different tenor in the two reviews we have achieved from you. We regret that you have the impression we are not willing to invest additional work for this study. Under point 4 of our reply from 3 August 2016 we followed your idea to compare your dataset with our study. Maybe one of your team can give us information on how to use the full VNF data files correctly and how to separate the flaring sources from other combustion sources (e.g. forest fires). With this data set, under consideration of the cloud correction, we will repeat our study.

M. Zhizhin provided us with the full VNF data for the relevant SWA countries in the time period of interest. The study has been repeated based on the new data and including the VIIRS cloud product.

Dear Mikhail Zhizhin (Referee, Geoscientific Model Development), thank you for your reviewer report from 14 October 2016. We have accounted for the comments and suggestions in the revised manuscript version. Please find our replies to the particular comments in the following.
We have uploaded a revised manuscript which also includes the revision regarding the comments and suggestions of the first reviewer.

Sincerely
Konrad Deetz and Bernhard Vogel

Referee comments:

In the paper a new method to model emissions from gas flaring is developed and validated on oil fields in Western Africa. The paper is a substantial contribution to the modeling science, and the approach is valid and motivating for further research. I have some comments on the presentation and details of the method which could be considered by the Authors before it is published.

0. I would recommend changing abbreviation VNP (VIIRS Nightfire Product) to commonly used VNF (simply VIIRS Nightfire) in the manuscript.
We agree on that and have changed the manuscript accordingly. For the general VIIRS Nightfire (including all combustion sources) we use the abbreviation "VNF" and for the extracted flaring information from VNF we use the abbreviation "VNF$_{flare}$".

1. Formula (1) derives gas flow rate from flare radiative heat and temperature measured from satellite. It is a basis of the proposed model. However, it is taken from Appendix of regulating document by the German Environmental protection agency. This is technical, not scientific source. The derivation of the formula is not provided neither in the paper under review, nor in the cited document. The cited document has no source for the formula either. It is important to derive the formula (1) or to provide a scientific reference.
We agree on that. TA-Luft is a technical document and the equation is not well introduced there. Equation 1 of the manuscript originates from VDI 3782, 1985: Dispersion of Air Pollutants in the Atmosphere, Determination of Plume rise, Verein Deutscher Ingenieure, VDI-Richtlinien 3782 Part 3, Equation 24, https://www.vdi.de/richtlinie/vdi_3782_blatt_3-ausbreitung_von_luftverunreinigungen_in_der_ atmosphaere_berechnung_der_abgasfahnenueberhoehung/ (accessed: October 17, 2016). We have changed the citation accordingly. Although VDI 3782 (1985) is also a technical document, the derivation of the equation becomes clear. The heat flow $M$ in MW is given by equation 1

$$M = c_p \, F \, (T_S - T_A), \tag{1}$$

where $F$ is the flow rate in m$^3$ s$^{-1}$, $c_p$ the mean specific heat capacity of the emissions, $T_S$ the source temperature and $T_A$ the ambient temperature. VDI 3782 (1985) provides a value of the mean specific heat capacity of

$$c_p = 1.36 \cdot 10^{-3} \, MW \, s \, m^{-3} \, K^{-1}$$

which is derived for a pit coal firing but VDI 3782 (1985) denotes, that this can be used for other flue gases as well since potential deviations are negligible. (An explicit $c_p$ value for gas flaring is not provided in the literature.) For the ambient temperature $T_A$ we use 298.15K as a fixed value, representative for the tropical region. Within a sensitivity study regarding the influence of $T_A$ on $F$ we have used the mean heat flow and the mean source temperature of all flares in TP15 and varied the ambient temperature between 293K and 303K, as a reasonable temperature range in the tropical regions. The resulting maximum difference in the heat flow is 0.0036 m3 s-1. Therefore we assume the errors using a fixed climatological value for the ambient temperature are negligible, but of course the user has to adapt the ambient temperature to the region he wants to apply the inventory. We have emphasized this in the manuscript. By using equation 1, the value for $c_p$ and for $T_A$, the flow rate $F$ in MW is given by:

$$F = M/\left(1.36 \cdot 10^{-3} \, (T_S - 298.15)\right). \tag{2}$$

2. Flare temperature used in the formula (1) is taken from instantaneous satellite measurement (VNF). It has a large variance depending on atmospheric conditions etc. I would recommend using mean flare temperature averaged over all cloud-free detections.

We see your point. This leads to a further source of uncertainty, because we cannot decide whether the spatial source temperature variations really results from the sources or from the atmospheric conditions. We think that this problem does not affect the climatological approach ($E_{clim}$ in the revised manuscript) because for every detected flare the source temperature already is averaged over the two-month period of TP14 or TP15 before we calculate the emissions. We assume that this is a compromise between robustness and keeping the spatial variability of the flaring. To allow for consistency we now also use these temporal averages of source temperature and radiant heat for the daily resolved inventories ($E_{obs}$ and $E_{com}$ in the revised manuscript). Therefore all three inventories have the same underlying emission field and the difference is just related to number of flares that are active at a certain day. For $E_{clim}$ all flares are active at once, for $E_{obs}$ only the actual observed flares are active and for $E_{com}$ the actual observed flares + the cloud covered flares (taken as active) are considered (com=combination). Nevertheless we have also included a further inventory in Tab. 5 that uses instantaneous input data to derive $E_{clim}$ (first calculating the emissions for every single observation and then averaging the emissions temporally). This is given as "$E_{clim}$, instant. input" and should allow further insight in the sensitivity/uncertainty.

3. The number 283 used in the formula (1) I believe stands for ambient air temperature at night? Is it a proper climatological value for Wester Africa?

Yes, the 283 refers to the ambient temperature. We agree that this value is not appropriate for the tropics. We have changed this value in the manuscript to 25°C (298.15K, also described in Comment 1 of this document). Owing to the change of the ambient temperature we have repeated our analysis to be consistent with this new value. The change from $T_A = 283K \, to \, 298.15K$ lead to a slight increase in the emissions (e.g. for Fig. 9b the spatially integrated SWA emissions of TP14 increase from 651 to 658 t h-1).

4. Comments 1-3 may result in a wider variance of the proposed model output, and the model sensitivity analysis should be presented.

Regarding the ambient temperature (Comment 3) we have presented the maximum uncertainty in the heat flow as 0.0036 m3 s-1, which is also described in the manuscript. For the mean heat capacity of the emission $c_p$ we do not have further information to assess the uncertainty. For considering the uncertainty in using temporal averages of source temperature and radiant heat instead of the instantaneous satellite observations, we have added a further emission inventory in Tab. 5 ("$E_{clim}$, instant. input", for TP14 and TP15).

5. The Authors have made a considerable effort to take into account cloud conditions which can mask flare observations from space. Why not to use only cloud-free observation days, and to count detected/not detected flare cases to derive mean radiative heat?

By using the postprocessed flaring data (VNF$_{flare}$ in the revised manuscript which includes also a cloud mask) instead of the "Flaring only" product, it is straightforward to separate the flares into the categories (a) "cloud-covered", (b) "cloud-free and inactive" and (c) "cloud-free and active". By assuming that the cloud-covered flares are active with their mean emission strength, we can estimate the daily emissions via the sum of (a) and (c). To use only the cloud-free observation days would be problematic because SWA is a region with very extensive cloud cover (on average approx. 70% in the flaring area).

I would like to acknowledge that the Authors provide software sources and input data used in the study as the paper supplement. It is helpful for reproduction and reuse of their science and model.

List of relevant changes

1. Use flaring information from "VIIRS Nightfire Nighttime Detection and Characterization of Combustion Sources" instead of the quick-look data from "VIIRS Nightfire (Flares Only version). The study has been repeated with this data.

2. More detailed description and derivation of Equation 1.

3. Add missing references.

4. Based on the availability of the VIIRS cloud product, the strategy of assessing the uncertainty of flares masked by clouds has been changed. Three categories were defined: (1) cloud-free and active, (2) cloud-free and inactive, (3) cloud-covered and assumed to be active. This leads to different emission inventories. We have defined two inventories in addition to the climatology ($E_{clim}$): (a) $E_{obs}$ which only consider the daily observations and (b) $E_{com}$ which is a combination of Eobs and the emission from the cloud-covered flares.

5. Assessment of a further source of uncertainty regarding instantaneous VIIRS observations vs. averaged VIIRS observations. This assessment lead to a further emission inventory: $E_{clim}$, instantaneous input.

6. Additionally, the sensitivity of the flow rate calculation in Eq. 1 towards the ambient temperature has been assessed.

7. Correction of the ambient temperature in Eq. 1 to consider tropical conditions. For consistency the study has been repeated.

8. Update of Tab. 5 based on the revised analysis.

[revised manuscript text omitted]

---

## Author Response (AR2)

Dear Axel Lauer (Topical Editor, Geoscientific Model Development), thank you for your report from 1 March 2017. We have accounted for the comments and suggestions in the revised manuscript version. Please find our replies to the particular comments in the following.

Sincerely
Konrad Deetz and Bernhard Vogel

Referee comments:
My following comments are based on the revised submission by the authors. The authors developed a method of estimating gaseous pollutant emissions from gas flaring for southern West Africa by incorporating various source data and theoretical equations. Gas flaring is not a global issue, but could be considerable at the regional scale, especially in southern West Africa where oil and gas production activities are substantial. I do think this study is very meaningful and innovative, however, the biggest flaw of this study comes from too many assumptions of the input parameters for estimating the emissions.

The key aim of our study is to describe gas flaring emission in southern West Africa (SWA) on a physical basis instead of using emission factors which hides all uncertainties in one number. The number of parameters indicates the complexity of the gas flaring emission. Assumptions are necessary in these cases where measurements were not available.

1) For instance, (1) the authors indicated that "IU14 remarked, that the reaction condition for flaring of $\eta \gg 0.5$ and $\delta > 0.9$ should be the norm in regions", but why $\eta$ was set to 0.8 and $\delta$ was set to 0.95?

The parameters combustion efficiency ($\eta$) and availability of combustion air ($\delta$) are strongly dependent on the type of flare and how the flaring process is handled. This can vary significantly from one site to another. For SWA we have no information about these parameters. Therefore we have on the one hand tried to isolate the parameter range according to literature values for general gas flaring (not specifically for SWA) and on the other hand conducted a sensitivity study to estimate the uncertainty (see Fig. 8a,b in the manuscript).

Regarding the combustion efficiency $\eta$ the studies IU14, Strosher (2000) and EPA (1985) were used. Based on these studies we have decided for $\eta = 0.8$. Regarding $\delta$ we have decided for $\delta = 0.95$ by following the remark of IU14 $\delta \geq 0.9$ and by assuming that the flaring conditions are not perfect in SWA and therefore that there is a deficiency in combustion air $\delta < 1.0$. Based on these limits we have decided for $\delta = 0.95$. We agree that the parameter selection of $\eta$ and $\delta$ was not motivated detailed enough. Therefore we have updated the relevant passage accordingly:

"The combustion efficiency $\eta$ and the availability of combustion air $\delta$ significantly depend on the flaring characteristics (e.g. available technique to steer the flaring process and how the staff takes care of the flaring procedure), which can vary significantly from one side to another. For SWA no information about these parameters is available. The parameter range at least was isolated according to literature values for gas flaring in general (not specifically for SWA). IU14 remarked, that the reaction condition for flaring of $\eta \gg 0.5$ and $\delta \geq 0.9$ should be the norm in regions, where the effective utilization of this gas is not available or not economically. Strosher (2000) indicates a combustion efficiency of solution gas at oil-field battery sites between 0.62 and 0.82, and 0.96 for flaring of natural gas in the open atmosphere under turbulent conditions. EPA (1985) shows combustion efficiencies between 0.982 and 1 for measurements on a flare screening facility. Based on these information the combustion efficiency η was set to 0.8. Regarding the availability of combustion air we on the one hand follow IU14 with $\delta \geq 0.9$ and on the other hand assume that the flaring conditions are not perfect in SWA, which means that there is a deficiency in combustion air $\delta < 1.0$. Therefore $\delta = 0.9$ was used for this study."

2) The specific heat capacity of associated petroleum gas should be highly dependent on its chemical composition. The application of a constant value is not appropriate.

The specific heat capacity $c_p$ (equation (2) in the manuscript) is that for the exhaust gas and not for the fuel gas. We agree that $c_p$ depends on the chemical composition of the fuel gas and that $c_p$ is spatiotemporally not constant. Since we use a spatiotemporal constant chemical composition of the fuel gas (based on Sonibare and Akeredolu (2004)) it is consistent also to use a constant $c_p$ value.

There is no information of $c_p$ of the fuel gas of the flares in SWA. Often waste gases from oil refineries are burned which can have other chemical compositions as the natural gas of this site has. Therefore even with a known chemical composition of the natural gas, the uncertainty in $c_p$ will stay.

Table 1 shows $c_p$ values for

I - used for this study, according to VDI 3782 (1985),

II-X - single components of the exhaust gas,

XI - the mixture of gases (II-X) according to the exhaust gas composition which was calculated from IU14 combustion equations.

**Tab. 1** – Comparison of the specific heat capacity $c_p$ for several gases.

|  |  | $c_p$ (J kg$^{-1}$ K$^{-1}$) | Flow rate $F$ (m$^3$ s$^{-1}$) relative to this study ($F \sim 1/cp$) |
|---|---|---|---|
| (I) | This study* | 1070.366 | 1 |
| (II) | Carbon dioxide | 846 | - |
| (III) | Carbon monoxide | 1040 | - |
| (IV) | Water | 1870 | - |
| (V) | Hydrogen | 14400 | - |
| (VI) | Oxygen | 912 | - |
| (VII) | Nitrogen | 1040 | - |
| (VIII) | Sulfur dioxide | 632 | - |
| (IX) | Nitrous monoxide | 1000.9 | - |
| (X) | Nitrous dioxide | 632 | - |
| (XI) | Mixture of gases** ($C_{p,mix} = \sum \frac{n_i}{n} c_{p,i}$) | 1110 | 0.965 |

*VDI 3782 (1985) using fuel gas density $\rho_f$

** flare mean of TP15

We have calculated (XI) for every flare of TP15 and the variation is below 2 J kg$^{-1}$ K$^{-1}$. When comparing (I) and (XI), the uncertainty when using (I) instead of (XI) is below 5%. Compared to the other sources of uncertainty (e.g. IU14 parameters, gauge pressure) this is negligible. We have added a comment in the manuscript:

"The value is consistent with the derived mean specific heat capacity for TP15 with an uncertainty below 5%."

3) The estimation of fuel gas density highly depends on the gauge pressure. It is indicated by the author the gauge pressure varied from 0 – 34kPa. By taking 0 and 34kPa as inputs, respectively, the range of estimated fuel gas densities could be 3 – 4 times different. However, the authors didn't explain why 34kPa was taken.

The gauge pressure is as uncertain as the selection of $\eta$ and $\delta$. As indicated in the manuscript, Bader et al. (2011) pointed out that the low-pressure single point flares, as the most common flare type for onshore facilities, operate at pressures below 10 psi (pressure above ambient pressure).

API (2007) remarks that most subsonic-flare seal drums operate in the range from 0-5 psi (pressure above ambient pressure). Therefore we have decided to use 5psi, the mean value between the limits 0 and 10 psi. The uncertainty due to the gauge pressure is shown in Fig. 8b for (0, 5 and 10 psi) and linked with that the influence on the fuel gas density. The uncertainty owing to the gauge pressure is part of the overall uncertainty estimation: $(^{+20}/_{-25}\%)$ .

4) Although the authors spent lengthy discussions in uncertainty analysis of different input parameters, the results of this study are not informative. As part of the field campaign DACCIWA, the mission should at least investigate some region-specific parameters of the fuel gas such as gauge pressure.

The assessment of the uncertainty the authors see as a key aspect of the study on hand. Without this analysis the study would be incomplete or would pretend certainty and robustness where the current level of knowledge has low confidence.

The DACCIWA field campaign took place from 1 June 2016 to 31 July 2016. This campaign includes (a) the three ground-based so called supersites Savé (Benin), Kumasi (Ghana) and Ile-Ife (Nigeria), which were measuring from 13 June to 31 July 2016, (b) radiosondes and (c) aircraft measurements from the three aircrafts: DLR Falcon, SAFIRE ATR-42 and the British Antarctic Survey (BAS) Twin Otter.

Regarding (a): Savé measured meteorological parameters and in addition the concentrations of ozone, nitrous monoxide, nitrous dioxide, carbon monoxide and isoprene as well as biogenic fluxes. Kumasi and Ile-Ife measured only meteorological parameters.

Regarding (c): The three research aircrafts conducted in total 50 missions (155 flight hours). This includes the three EUFAR (European Facility for Airborne Research) missions OLACTA (Observing the Low-level Atmospheric Circulation in Tropical Atlantic, 10 flight hours), MICWA (Mid-level Inversions and Cloudiness in SWA, 10 flight hours) and APSOWA (Atmosphere Pollution from Shipping and Oil platforms in West Africa, 10 flight hours).

The following objectives were targeted for the aircrafts: characterization of stratus clouds and their interaction with aerosol; quantification of city emissions from Lomé, Accra, Abidjan, Cotonou and Kumasi; characterization of power plant emissions, oil and gas flaring; air pollution from shipping; effects of clouds and radiation; interaction between the land-sea breeze and clouds; measurement of BVOCs and the examination of dust and biomass burning aerosol.

The aircrafts measured meteorological parameters, trace gas and aerosol concentrations as well as cloud droplet and aerosol size distributions and the aerosol composition.

Based on this short overview of the DACCIWA field campaign, the following remarks have to be made according to the referee comment (4):

- The air pollution from flaring was not a key aspect of DACCIWA. It is just one of the sources of air pollution which contributes to the atmospheric composition of SWA.

- The supersites do not contribute to the analysis of the flaring emissions.

- The aircrafts were not allowed to enter the Nigerian airspace and therefore the extended onshore and offshore oil fields in the Niger Delta have not been observed (see Fig. 1 below). This means the only flares reachable in SWA were the sporadic offshore flares south of Ivory Coast and Ghana (see Fig. 2 below).

- Out of the 50 aircraft missions only 2 were explicitly dedicated to flaring.

- Aircraft observation can provide the concentration of trace gases and aerosols from a source but they neither can provide information about the emissions nor about flare specific information like gauge pressure or specific heat capacity.

- To get information about the characteristics of the flare stack, the flaring process and the composition of the fuel gas, a detailed and long-term study directly at the flaring sites would have been necessary. We cannot expect that the gauge pressure is spatiotemporally constant. It is very likely that it significantly varies between several sites and during different working processes at the flaring site. These observations (long-term or short-term) could neither be handled within DACCIWA nor would be granted by the authorities or the oil companies. The oil companies do not cooperate in providing these parameters since flaring is a controversial topic in society and politics.

- The EUFAR mission APSOWA aims to characterize gaseous and particulate pollutants emitted by shipping and oil and gas extraction platforms off the coast of West Africa. These observations shows flare related peaks in sulfur dioxide (~8 ppb), carbon dioxide (~175 ppb) and nitrous dioxide (~7 ppb). However, aircraft observations are not constructive in deriving the parameters needed for the parameterization which is presented in this study, even if a sufficient number of flights (statistics in terms of different flares and number of observations) would have been dedicated to analyze the flaring.

- The very sporadic measurements of flaring concentrations are also not appropriate for direct evaluation of the output of the flaring parameterization because the emissions have to be transformed to concentrations. Therefore a link of an atmospheric dispersion model with the emission parameterization is necessary to be comparable which the aircraft observations. This brings further uncertainty into the intercomparison, especially if only a very small number of aircraft detections of flaring pollution are available. We have added a passage in section 5 to denote this problem:

"Gas flaring is just one of the sources of air pollution in SWA and therefore the DACCIWA field campaign in June-July flaring cannot solely focus on flaring. To provide detailed measurements of the flaring characteristics would go beyond the scope of DACCIWA. However, within the DACCIWA aircraft campaign, the EUFAR (European Facility for Airborne Research) mission APSOWA (Atmosphere Pollution from Shipping and Oil platforms in West Africa) was conducted to characterize gaseous and particulate pollutants emitted by shipping and oil and gas extraction platforms off the coast of West Africa. The authors hope that the results of APSOWA bring further insight in the characteristics of gas flaring in SWA."

[Figure]

**Fig. 1.** – Overview of flights conducted by the three research aircrafts during the DACCIWA field campaign. The DLR Falcon is denoted in green, the SAFIRE ATR-42 in red and the BAS Twin Otter in blue. Nigeria (east of Benin) shows no aircraft observations. (This figure is not part of the manuscript.)

[Figure]

**Fig. 2.** – Location of gas flaring in June-July 2014 (red), June-July 2015 (green) and both periods (grey). The countries in the north are Ivory Coast, Ghana, Togo, Benin and Nigeria (from west to east). (This figure is also part of the manuscript (Fig. 1).)

5) Most importantly, the emissions should be evaluated by the model as the campaign should have a suite of measurement data. Without this validation, this manuscript seems not coincide with scope of GMD. Therefore, I suggest a resubmission by comprehensively evaluating the emissions created in this study.

This refers to the referee comment (4). DACCIWA has not provided data which allows a comprehensive evaluation. In the manuscript we therefore have conducted a detailed evaluation against existing flaring emission inventories. The aim of our study was to shed light on flaring as a side topic of DACCIWA, including all the uncertainties which are linked to the description of gas flaring emission. When starting with this study, very low information was available for the flaring in that region. Within our study we bring the information together which are available (e.g. the gas composition of Sonibare and Akeredolu (2004) and the space-borne observations of VNP (flare location and flare activity) to describe the flaring emissions on a physical basis using IU14. This can be seen as a significant step forward in comparison to the description via emission factors. We do not deny the uncertainties within the parameterization; on the contrary we disassemble the uncertainties to lay them open, instead of hiding them in the description of emission factors.

To further raise the knowledge about the characteristics and the amount of gas flaring in that region, the will of the local politicians together with the cooperation of the oil producing industry is necessary.

We have described our method in detail, evaluated it against existing inventories and made the source code free available for the reader/user in terms of reproducibility. Within the aims and scope of GMD our study is located in "development and technical papers, describing developments such as new parameterizations of technical aspects of running models such as the reproducibility of results".

[revised manuscript text omitted]